# Early Life Stress (ELS) Effects on Fetal and Adult Bone Development

**DOI:** 10.3390/children10010102

**Published:** 2023-01-03

**Authors:** Xena Giada Pappalardo, Gianluca Testa, Rosalia Pellitteri, Paola Dell’Albani, Margherita Rodolico, Vito Pavone, Enrico Parano

**Affiliations:** 1Institute for Biomedical Research and Innovation, Italian National Research Council, 95123 Catania, Italy; 2Department of General Surgery and Medical Surgical Specialties, Section of Orthopaedics and Traumatology, Hospital Policlinico “Rodolico-San Marco”, University of Catania, 95123 Catania, Italy

**Keywords:** bone development, child abuse and maltreatment, early life stress (ELS), epigenetics, genetics, maternal stress, osteoporosis, perinatal stress, postnatal stress, prenatal stress

## Abstract

Early life stress (ELS) refers to harmful environmental events (i.e., poor maternal health, metabolic restraint, childhood trauma) occurring during the prenatal and/or postnatal period, which may cause the ‘epigenetic corruption’ of cellular and molecular signaling of mental and physical development. While the impact of ELS in a wide range of human diseases has been confirmed, the ELS susceptibility to bone diseases has been poorly explored. In this review, to understand the potential mediating pathways of ELS in bone diseases, PRISMA criteria were used to analyze different stress protocols in mammal models and the effects elicited in dams and their progeny. Data collected, despite the methodological heterogeneity, show that ELS interferes with fetal bone formation, also revealing that the stress type and affected developmental phase may influence the variety and severity of bone anomalies. Interestingly, these findings highlight the maternal and fetal ability to buffer stress, establishing a new role for the placenta in minimizing ELS perturbations. The functional link between ELS and bone impairments will boost future investigations on maternal stress transmission to the fetus and, parallelly, help the assessment of catch-up mechanisms of skeleton adaptations from the cascading ELS effects.

## 1. Introduction

Early life stress (ELS) is a wide concept including several types of aversive events, which may occur in two distinct moments of life, during the prenatal and the postnatal period, or may involve both, defined as perinatal stress. Prenatal stress coincides with intrauterine life, and it is often referred to as maternal stress which affects fetal health, while postnatal stress indicates the timeframe of development spanning until adult age [1,2,3]. The negative effects of prenatal stress do not end after birth but can persist based on the postnatal environment with or without stress. Furthermore, the impact of postnatal stress can be long-lasting and can be compounded by further challenging events encountered during growth [4]. Each of the two forms may predispose the organism to many adult diseases, albeit in worst-case scenarios, and an individual can also experience both across the lifespan (the two hits hypothesis), increasing the risk for later health consequences [4], as stated by the well-known Barker’s theory of developmental origins of health and disease (DOHaD) [2].

The relevance of ELS effects in mediating pathological pathways has been confirmed in humans by short- and long-term biological changes of organs and tissues, which implicate both structural and functional alterations of underlying cellular and molecular mechanisms of genome regulation [5]. More explicitly, the plasticity of the human genome programs adequate responses to multiple cellular and environmental stimuli, ensuring the survival of the organism through different adaptive strategies.

The adaptive process involves a great number of developmental trajectories established by the complex interplay of genetic (or constitutive) and environmental (epigenetic) factors [6]. The summation of various factors influencing the decision-making trajectories of development leads to different phenotypic traits, which are not always the most performant ones. Adverse signals are known to completely destabilize genome regulation or may influence the expression of genes, giving rise to more adaptive phenotypes able to survive to certain environmental conditions to which the organism is subjected.

However, in the absence of such needs or in other surroundings, the phenotypes present with maladaptive and pathological traits. The most studied example is the genome response to Dutch famine (the ‘Hunger Winter’) during the Second World War, also called the ‘thrifty phenotype hypothesis’ by Barker and Hales [7], since prenatal exposure to the undernourished state has been associated with an increased risk of developing type 2 diabetes. The food shortage induced the beta cells and the islets of Langerhans to lower the insulin levels, but the adaptive role of being metabolically thrifty was lost with the change of nutritional habits by the increased consumption of food. Notably, the transmission of the metabolic deregulation has been detected in subsequent generations, highlighting the role of long-lasting epigenetic changes [8]. There are several important findings sustaining the early life programming of adult diseases caused by ELS, among which are cardiovascular disorders [9], age-related disorders [10], and depression [11]. In light of this evidence, the genome exposure to numerous perpetrated and deleterious signals received during evolutionary phases more sensitive to changes in body function and formation may lead to alteration of genomic regulation, unless other factors act as a shield from the harmful cascading effects of ELS and positively modulate the genomic responses [5]. At the molecular level, aside from the classical operational mode of genetic mutations, which affect the nucleotide sequence, environmental stressors may change gene regulation with immediate, intermediate, and late effects through so-called epigenetic factors [5]. In fact, these factors are involved in genome plasticity through a ‘switch on-off’ mechanism to activate or silence gene expression by the addition or the removal of specific chemical signals in the DNA sequence (i.e., DNA methylation) or in the chromatin (i.e., histone modifications). Certain epigenetic tags induce temporary and reversible adaptations, eliciting a considerable interest in exploring their modifications as potential therapeutic targets for a wide number of complex diseases [12], including a novel bone-active medication for osteoporosis [13].

The musculoskeletal tissue, as well as the nervous system, are the most responsive tissues to various environmental stimuli in the fetus, from the daily stress of pregnancy to maternal nutritional status, as well as smoking and alcohol and drug use [14], in addition to traumatic experiences such as intimate partner violence [15]; all of these ELS types affect the physical remodeling and molecular signaling of bone homeostasis. The intrinsic versatility of bone physiology provides great resistance and reactivity to several negative conditions of pregnancy, such as unhealthy feeding and care in pre- and postnatal life, but it also reveals to be one of the most exposed tissues to the ELS load during fetal and childhood development [16]. The impact of ELS on bone development has been not extensively investigated in humans, although acute and chronic stress has been documented to be associated with intrauterine growth restriction (IUGR), postnatal growth retardation, and reduced bone mineral density (BMD) [17,18]. Moreover, there is evidence linking depression with bone fragility and osteoporosis in adults [18,19,20,21]. However, the current knowledge about the underlying mechanisms between ELS, stress-responsive genes, and detrimental changes in bone metabolism has been limited for several reasons. First is that, although childhood maltreatment is one of the major risk factors of ELS, a great number of victims are hidden in the home and often underreported [22]; second, it is rarely diagnosed due to inadequate training of pediatricians [23]; third, the phenomenon is often retrospectively investigated due to social and ethical requirements [24]. Therefore, animal models have been used to overcome these historical problems, allowing the investigation of underlying mechanisms in a highly controlled environment.

The present review provides an overview of pre-, post-, and perinatal studies of ELS in non-human mammals in order to construct the first analysis of ELS as a risk factor for short- and long-effects on bone pathologies.

## 2. Materials and Methods

The literature review was performed according to the guidelines of the Preferred Reporting Items for Systematic Reviews and Meta-Analyses (PRISMA) [25] (Figure 1) (Appendix A). Primary research and reviews from PubMed and Google Scholar were selected from 2000 to September 2022. The online search was organized in two parts: the first part focused on clinical evidence of the association between bone diseases and stress in early life, while the second one included genetic and epigenetic correlations.

Relevant articles describing the effects of prenatal stress on fetal and child bone development were manually examined, and genotypic, phenotypic, and functional data were extracted for analysis. After the removal of duplicate records, the main results of this search were included in the present reference list and summarized in Table 1.

## 3. Results

ELS affects all systems of the body, leading to several mental and physical diseases in adults [1]. However, among these, there is poor evidence supporting the influence of ELS on musculoskeletal development and susceptibility to bone diseases. The main limitation is linked to the general assumption of a narrow timing window of exposure to ELS maintained until now in pediatric orthopedics by most studies focused on bone injuries and fractures caused by abusive episodes during infancy and adolescence [40,41].

The relevance of an early diagnosis of child abuse and maltreatment has almost entirely eclipsed the recognition of warning signs of other ELS types threatening the skeletal health of newborns in the womb. Another restraint is the lack of a methodological approach to identify and monitor the risk of maternal factors (i.e., diet, behavior, and stress) that may negatively influence fetal bone mass and early growth, and newborn fractures are still debated [42,43].

In the following paragraphs, the analysis of the collected evidence will be discussed, showing the synthesis of in vivo models for the evaluation of ELS impact on bone development and the genetic and epigenetic contribution to the regulation of bone homeostasis in pre-, post-, and perinatal life.

### 3.1. ELS Impact on Bone Development

The restricted number of stress simulation studies, albeit variegated for laboratory animals, ELS type (pre-, post-, and perinatal), and the stress protocol (experimental setting and time points) are summarized below, highlighting their respective strengths and vulnerabilities (Table 1).

To address the ELS pattern on bone development within a unitary framework of data, our analysis excluded findings from (i) models of non-mammal animals (i.e., bird and fish), which have no analogies with the potential perturbations of a stable and controlled environment as the womb; (ii) studies of maternal stress not focused on the offspring’s bone health, which measure the birth weight of the offspring as one of the main features of the general developmental status indirectly providing information of incomplete ossification of the pup’s skeleton.

#### 3.1.1. Prenatal Stress Models

Mouse

The study of Lee et al. [26] elucidated the impact of maternal stress on embryos and fetuses at different gestational stages. Pregnant mice were subjected to daily 12 h restraint stress, taped in the supine position on a plastic board, on gestational days (GD) 1–4, 5–8, 9–12, and 13–16, respectively. During the daily restraint for 4 days, the maternal body weights markedly decreased. Although the body weights improved gradually after stress, the recovery was not full until the final stage of pregnancy. Restraint stress caused growth retardation of the fetuses. Although the preceding (GD1-4) and concurrent (GD5-8) stresses did not affect embryonic implantation, restraint stress on GD9-12 caused cleft palate. Vertebral abnormalities, mainly bipartite ossification, were observed only in animals stressed on GD5-8, while abnormalities of sternebrae and asymmetric or bipartite ossification were enhanced at all gestational stages. These results suggest that intensive restraint stress influences maternal body weight, consequently body size variations and increased mortality of embryos and fetuses, in addition to gestational-stage-specific ventricular dilatation, cleft palate, and sternal abnormalities.

Kim et al. [27] used two different doses of dexamethasone (Dex) (1 mg/kg or 10 mg/kg maternal body weight per day) administered intraperitoneally at GD7-9 in pregnant mice which were sacrificed at GD18. Seven out of eighteen (39%) embryos treated with a high dose showed axial skeletal abnormalities in either the T13 or L1 vertebrae. In addition, the examination of the sternum revealed that the xiphoid process, the protrusive triangular part of the lower end of the sternum, was bent more outward or inward in Dex-treated embryos compared to the controls. It was also noted that the angle divergence between the sternum and the protruding end of the xiphoid process was bigger in female embryos than in male embryos. This finding may be consistent with the gender difference in the effect of prenatal glucocorticoid exposure or stress [44].

Choe et al. [28] carried out the first integrative approach to chronic maternal stress, detecting phenotypic and genomic changes through the assessment of body weights and skeletal variations, also including the transcriptomic and epigenetic analysis of developing limbs from control-matched and stressed pregnant mice. Dams were daily immobilized for 6 h from GD8 and then sacrificed. According to this experimental model, a significant decrease in fetal body weight, as well as delays in several developmental events, such as somite numbers and limb bud formation, and regression of interdigital webbing were observed in the stressed fetuses. Molecular findings revealed the impairment of the fetal bone transcriptional program, identifying both a deregulated set of developmental-specific genes and an altered global gene expression pattern of limb development. In fact, they found in the limbs of maternally stressed fetuses the suppression of the Igf1 (insulin-like growth factor 1) gene, normally expressed during fetal limb development, as well as the Acta1 (actin alpha 1) gene, a component of mature skeletal muscle, associated with delayed features of limb developments. Additionally, they measured the lower expression of the Aldh1a2 (aldehyde dehydrogenase 1 family member a2) gene involved in multiple developmental pathways (i.e., limb bud initiation, proximodistal outgrowth, apoptosis of interdigital tissue, and chondrogenesis) and also the delayed expression of the Fgf8 (fibroblast growth factor 8) gene, a well-known marker of the apical ectodermal ridge related to limb buds [45]. The comparative pathway analysis showed that the most ranked biological processes of upregulated transcripts were transport, signal transduction, and protein metabolism, whereas downregulated transcripts were associated with organelle organization, transport, protein metabolism, and development. Of note, authors evaluated changes in the methylation profile of the promoters of differentially expressed genes. Although no significant methylation changes were detected using the real-time PCR assay based on the methylation-specific restriction enzyme analysis, it is possible that other higher-resolution techniques and extending the promoter region of analysis can improve the identification of differences of methylation level [46].

Azuma et al. [29] evaluated if certain behavioral strategies for coping with stress enacted by mothers may attenuate harmful consequences of ELS in the newborns, for instance, chewing, which is a small action implicating many benefits, such as stress relief and cognitive improvement. To distinguish any biological differences in stress management, the experimental protocol entailed the detention of pregnant mice at GD12 in a ventilated restraint tube for 45 min 3 times a day until delivery, allowing a group of stressed mice to chew on a wooden stick during the restraint stress period. Authors assessed the bone response of 5-month-old male offspring, comparing control, stressed, and stress/chewing groups using quantitative micro-computed tomography, bone histomorphometry, and biochemical markers. Their findings showed that maternal chewing during prenatal stress appeared to be effective for preventing lower bone mass in the adult offspring compared to pups born from non-chewing stressed mice. A significant decrease was found in trabecular bone mass in both vertebrae and the distal femur of the offspring of non-chewing stressed mice. Whereas the lower trabecular bone volume and bone microstructural deterioration were improved in the chewing group, the inspected parameters of both groups were significantly lower than the control group. More strikingly, no significant changes in body weight were observed in neonatal pups or in offspring at 5 months of age in any of the three groups, contrary to evidence that the birth weight is the first health index affected by ELS. This discrepancy may be linked to the type of stress protocol and the selection of time window used for ELS investigation, also suggesting that mechanisms of intrauterine growth retardation should be further elucidated [27].

This task has been generally explored by ecological studies identifying the pattern of developmental perturbations (with the use of parameters like fluctuating asymmetry and frequency of phenodeviants) on the morphological variations, including the cranial or skeletal shape [47]. Very little research is available on the developmental instability of skeletal pattern induced by adverse intrauterine and postnatal environment.

In this regard, Gonzales et al. [30] used a mouse model of maternal caloric restriction until GD17-18 to explore the influence of nutritional imbalance in the fluctuating asymmetry of cranial structures, attributed as an index for inferring stress among individuals. According to these results, a significant reduction in skull size in mice born under maternal nutritional stress was detected, demonstrating that the prenatal perturbation induced changes in the spatial pattern of fluctuating asymmetry of the skull, but not in the magnitude.

Rat

Among laboratory animals, models on rats have provided more insightful findings to understand the long-term effects of ELS and the risk of developing disease in adult phenotypes, such as osteoporosis, as reviewed in [48].

Amugongo et al. [32] carried out a twelve-month experimental protocol to verify whether the stress on a pregnant female had a significant negative impact on the offspring’s weight gain and skeletal development, despite the offspring being kept under stress-free postnatal conditions. The protocol was based on immobilization stress induced in pregnant mothers at various times of gestation days as follows: GD1-7 (Group 1), GD8-14 (Group 2), and GD15-21 (Group 3), plus an unstressed control group. The immobilization stress was daily administered in three sessions lasting 45 min. Maternal cortisol hormone, food intake, and weight gain were monitored during pregnancy. Pups were raised under normal laboratory conditions and sacrificed at ages: 4, 8, 12, and 16 weeks to determine the effect of prenatal stress. Cortisol hormone levels in controls were lower than those of stressed animals. Stressed dams consumed 12.5% less food per day compared to controls. Animals in Group 1 and Group 2 gained less weight during pregnancy but had larger litters than did Group 3 or the control group. Offspring born to Group 3 were heavier compared to all other groups. Group 3 offspring had a higher rate of bone formation. In conclusion, stress during pregnancy resulted in increased cortisol and reduced food intake in mothers, but faster growth and higher weight gain in offspring compared to controls. This finding was coherent with catch-up growth features of prenatally stressed animals, in which physiological and metabolic adaptations are programmed for survival in the adverse uterine environment and in a postnatal environment presumably similar to the uterine condition. Instead, it was agreed that the mismatch of over-adapted metabolism with pleasant and healthy postnatal environment makes them maladapted and may predispose them to metabolic disorders, like diabetes, at a later stage in life [7,49]. In this study, however, the presumed impact of a metabolic shift was not evaluated in the offspring experiencing a stress-free living condition.

The analysis of Swolin-Eide et al. [33] investigated whether the exposure to Dex had any effect on skeletal growth and/or BMD in adult rat offspring. Pregnant rats were given intramuscular injections of either Dex (100 micro g/kg) or vehicle (physiological saline) on GD9, 11, and 13, three specific time points associated with the sensitive period of early fetal brain development. Some pups of each gender and group were sacrificed at 6 weeks of age, while the rest of the offspring were kept until 10 weeks of age for male rats and 12 weeks of age for female rats. Dex-exposed male but not female rat offspring showed transient increases in crown–rump length and tibia and femur lengths at 3–6 weeks of age. In contrast, the cortical bone dimensions were altered in 12-week-old female but not male Dex-exposed offspring. The areal bone mineral densities of the long bones and the spine were unchanged in both male and female Dex-exposed offspring. The collected evidence established that prenatal Dex exposure affected skeletal growth in a gender-specific manner, but not the mineralization of bones.

A more recent contribution to elucidate gender differences in skeletal deformities is the ‘double-hit model’ of stress provided by Anevska et al. [35]. This work is a continuation of their previous publications [34,50,51,52,53] about the evaluation of fetal outcomes of intergenerational maternal stress through the analysis of the body weight and bone markers in the offspring (F1) born to mothers (F0) with pregnancy complication and then in the pregnant rats of F1 exposed to stress and in their progeny (F2). The pregnancy complication of the initial generation (F0) females was imposed at GD18 by the procedure of bilateral uterine vessel ligation in order to mimic uteroplacental insufficiency, which typically occurs during the third trimester of human pregnancies, associated with fetal growth restriction and reduced fetal skeletal mineralization [52]. Uteroplacental insufficiency resulted in low birth weight observed in the first generation (F1). As such, several anomalies in bone density and geometry were detected in the restricted females F1, such as shorter femurs and reduced trabecular and cortical bone mineral content (BMC), but once pregnant, their bone deficits were restored and not passed onto the subsequent generation (F2). This finding supports the activation of positive adaptations of pregnancy, which may have prevented the transmission of bone defects to F2 offspring [34]. The stress protocol tested in F1 pregnant growth-restricted females involved some physiological measurements (tail cuff blood pressure, glucose tolerance test, metabolic cage experiment) at GD18-19 in order to elicit a maternal moderate stress response with increased corticosterone [54].

Ultimately, evidence collected in [35] provided that maternal stress during pregnancy reduced birth weight in both F2 male and female offspring; however, postnatal and adult body weights did not change among the groups for either gender. The decreased birth weight may be considered a result of exposure to elevated concentrations of maternal corticosterone during late pregnancy, and the bone strength reduction in F1 females is associated with an increased risk of fracture [54].

Rabbit

Bots et al. [37,38] investigated the role of different levels of maternal stress in developmental instability of fetal skeletal abnormalities. In the first study, authors analyzed the linear patterns of the fluctuating asymmetry of the limbs [37], while in the second one, they also examined the frequency of phenodeviants [38]. Although both studies used gravid rabbits with a control, exposed to toxic treatments from GD6-19 and sacrificed on day 28, just before natural delivery, two different toxicological frameworks were reproduced. Specifically, in [37] the toxic compound was an antiprotozoal agent daily administered in three dose groups (80, 320, and 1280 mg/kg), mainly affecting fetal condition more than maternal, while in [38] hydroxy-propyl methylcellulose was used, a non-fermentable semi-synthetic dietary fiber, at different levels (100, 500, and 1500 mg/kg), known to reduce food consumption and cause weight loss in dams without toxic effects in the offspring. In the first study, authors found a higher fluctuating asymmetry of the hind limbs in the low treatment than in the control, detecting abnormalities in the high dose only [37], which was coherent with the so-called hypothesis of the “early warning system”, which states that fluctuating asymmetry could serve as a predictor of changes in fitness and health [55]. Examining the findings of the second study [38], the food shortage and weight loss of pregnant rabbits given the medium and high dose resulted in lowered fetal weight and transient alterations in ossification; this developmental delay is associated with growth retardation, confirming the primary origin of prenatal stress [56]. Moreover, the stress procedure with hydroxy-propyl methylcellulose, rather than with injecting corticosteroids, established a more realistic induction of maternal stress hormones and determined direct effects on fetal outcomes, since the administrated compound affected maternal food restriction, which increases the baseline glucocorticoids and the transplacental transfer of maternal cortisol [56]. Interestingly, in both studies no dose–response curve was observed of the potential relationship between developmental instability and maternal stress, suggesting the role of buffering mechanisms of the placenta [57].

Pig

As described in Choe et al. [28] with the chewing strategy in pregnant mice, Sliwa et al. [39] assessed the capability of the biological response of maternal stress to prevent fetal bone changes when, during pregnancy, the toxic effect of Dex is balanced with a safe and stimulant molecule like alpha-ketoglutarate (AKG). Here, authors tested the single or simultaneous supplementation of Dex as an inhibitor of the synthesis of collagen and the bone matrix and AKG as a metabolic inducer of growth. Previous studies have suggested that corticosteroids impair the linear growth of bone due to mechanisms of aberrant organization of the growth plate and collagen matrix [58]. To test this, four different groups of sows were treated during the last 24 GD with the administration of: (1) oral AKG (0.4 g/kg BW/day dosage); (2) intramuscular Dex (3 mg/sow/48 h dosage); (3) both AKG and Dex; and (4) intramuscular physiological saline in the controls. After birth, the piglets were sacrificed to examine two main bone formation markers, the alkaline phosphatase (AP) activity and osteocalcin (OC) level, as well as the mechanical parameter values of BMD and BMC of humeri.

First of all, the selection of the late gestational time for the execution of treatment was insightful for observing any impairment of the mineralization processes in the fetus that could not be detected before the second trimester according to the known pattern of fetal skeletal formation strictly dependent on the maternal metabolism [59]. Furthermore, this model demonstrated the prenatal influence of maternal administration of Dex in the newborn piglets characterized by reduced BMD and BMC of the humeri and the AP and OC serum level, while the simultaneous administration of Dex with AKG to pregnant sows increased all the investigated bone mechanical and biochemical values in comparison with other experimental groups. Of note, the administration of AKG only failed to enhance bone turnover, as AKG-treated pregnant sows delivered piglets with heavier weight but with less mineralized humeri than the control group, suggesting that this is not enough on its own to promote development and skeletal mineralization.

According to these results, the use of AKG as a diet supplement can improve growth through the activation of gluconeogenesis and ammoniagenesis in the fetus, while its role is not clear in the rescue mechanism from bone loss induced by Dex when double administered. Glucocorticoids are involved with other biological components in several different signal transduction mechanisms, however poorly identified [60]. Recent evidence has elucidated the metabolic pathway linking AKG with the prevention of bone loss and skeletal muscle protein degradation caused by the steroid medications [61].

#### 3.1.2. Postnatal Stress Model

Mouse

There are hardly any studies on postnatal impact. Only recently, translational research in humans was conducted by Wuertz-Kozak et al. [31] to investigate the tripartite effects of ELS on bone, endocrine, and nervous system development, closely connected to each other [62]. Moreover, this study first highlighted that depressed people who experienced episodes of ELS such as childhood abuse and neglect (ELS-depressive patients) compared with depressive patients without ELS were associated with a higher risk for reduced BMD, osteoporosis, and bone fractures [21]. The results of bone microarchitecture and metabolic and neuronal stress markers from the mouse model of ELS, named MSUS (unpredictable maternal separation and unpredictable maternal stress) [63] were compared with a sample of depressive patients with or without ELS by analyzing BMD and metabolic changes in serum (i.e., osteocalcin, procollagen type 1 N-terminal propeptide, PINP, c-terminal telopeptide of type I collagen, CTX-I). The experimental MSUS paradigm performs the so-called unpredictable maternal separation and maternal stress during early life to simulate childhood maltreatment and induce long-lasting health effects, such as neuropsychiatric and behavioral problems [63]. According to this, on postnatal day 1 (PND1), pups were separated during the dark cycle from their mother for 3 h per day from PND1 until PND14. In addition, mothers were subjected to a forced swim in cold water (18 °C for 5 min) or restraint in a plastic tube (20 min) at unpredictable times during the 3 h of separation. MSUS pups and controls were sacrificed at the age of 8–10 months under stress-free conditions. Their findings revealed that postnatal stress in the MSUS model led to low birth weight, altered bone innervation, and decreased expression of some neurogenic and osteogenic mediators of bone metabolism (i.e., nerve growth factor, *NGF;* neuropeptide Y receptor 1, *NPYR1;* vasoactive intestinal peptide receptor 1, *VIPR1*; tachykinin receptor 1, *TACR1*), but did not affect the expression of bone markers within the bone or bone microarchitecture. Moreover, no gender differences were observed in the susceptibility to bone alterations and/or sensitivity to stress. This response pattern was suggestive of a ‘catabolic shift’ associated with a higher rate of bone turnover and lower bone healing capacities, as a consequence of ELS, implicating the downregulation of the bone genetic program, reduced bone remodeling, and long-term destabilization. A reduction in BMD was also evident in depressive patients with ELS experiences of childhood abuse and neglect, compared to depressive patients without ELS. Interestingly, a different trend in bone metabolism was observed within the ELS-depressive patient group.

In fact, depressive patients with childhood abuse showed an increased bone metabolism, not observed in depressive patients with childhood neglect. These data suggested that the changes in bone serum markers and bone health (fractures, aging, etc.) may also be dependent on the type, severity, duration, and reiteration of ELS exposure, as well as accumulating damage over time (allostatic load), and should be interpreted in a more specific manner. Lastly, the results obtained in the MSUS model were only partially comparable with human data, since the patterns of stress emulated acute effects in mice and chronic effects in humans.

#### 3.1.3. Perinatal Stress Model

Rat

Dancause et al. [36] reported how prenatal (GD14–21), postnatal (PND2–9), and both pre- and postnatal stress effect long bone length. The prenatal procedure to induce maternal stress involved (in daily order): light from 2000 h–0800 h, housing in a wire-mesh-floored cage (24 h), food deprivation (12 h), tilting the cage 45° (6 h), exposure to strobe light (1 h), forced swim (10 min), restraint (30 min), and wet bedding (10 h). The postpartum stress schedule was similar, including (in daily order): strobe light, wet bedding, wire-mesh-floored cage, food deprivation, restraint, male intruder (5 min), forced swim, and housing in a small cage (designed for mice, 24 h). Offspring were exposed to all stressors instead of restraint and swim stress. According to their results, the tibia length was reduced in both male and female offspring in the three treatment groups (pre-, post-, and perinatal stress), while the femur length was only different among the males compared with controls. However, these findings seem to be in contrast to Anevska et al. [35], in which variations in the femur length after maternal stress were not found either in F2 male or female offspring.

## 4. Discussion

The present review provided a map of the available research literature on the detrimental effects of ELS on bone tissue during the fetal and postnatal development. After describing a preliminary assessment of potential scopes of pre-, post-, and perinatal studies in animal models, we examined the shift of benefits and harms of stress schedule (experimental, rational, and protocol). In the human population, studying ELS is hardly accessible since it is a frequent phenomenon associated with domestic violence against women during and after pregnancy, threatening the life of the fetus and then of the child, who is often the witness of the abuse of the mother or, in turn, being abused [64]. Since domestic violence and child maltreatment are often hidden crimes, diagnosis is rare and underestimated due to the inadequate formation of physicians [23] and the lack of effective and technologically advanced protective measures [41,65]. Most evidence of ELS-related diseases is based on retrospective studies, prevalently including neuropsychological, autoinflammatory, and metabolic disorders, whereas the risk for bone health has not been extensively investigated [19]. This implies that all data in the ELS regulation of bone development come from longitudinal case-control studies of animal models. Despite the heterogeneity of animal models and procedures mimicking human ELS, our analysis found a common denominator able to establish that both maternal and postnatal stress may lead to low birth weight and abnormalities in bone markers and skeletal growth (Table 1). In fact, differences in outcome assessment depend on several factors, such as the time period of stress, whether in utero during various pregnancy stages (early, mid, and late) or during the early development of the offspring (postnatal day, week, month) [3], as well as the pharmacological and physiological variety of the stress procedure, which influences the number and type of phenotypic effects―measurable changes in skeletal anomalies, gender differences, maternal catch-up mechanisms, and litter stress exposure in fetal bone formation processes. Divergent timelines of experimental evidence performed during the last trimester of gestation [30,35,39], contrary to those carried out during the first trimester [27,29,33,38,39,44,45] or involving both stages [32,63], made the comparison of ELS data indicative rather than scientific. This means that it was difficult to find valid similarities with human pregnancy for which the first trimester is believed to be the most stress-sensitive and riskiest period for abortion [3]. It is also very plausible that, depending on the magnitude (i.e., severity of discomfort, allostatic load) and the extension of the stress exposure (i.e., acute or chronic), there is a stress tolerance window that could be modulated in the dams and litter to activate several catch-up mechanisms through placental and postnatal adaptions [32,38].

Moreover, each prenatal and postnatal developmental stage would also represent a temporal segment of the individual tolerance window that might be sex-dependent as reported by [33,34,35,44], since the neuroendocrine hormones and hypothalamic pituitary adrenal axis (HPA) mediate the different susceptibility to bone alterations and/or sensitivity to stress [66]. The mitigation and adaptation schedule of bone morphogenetic protein pathways to the prenatal stress effects was explored by means of antistress chewing and intake of the restorative supplement AKG, respectively, in pregnant mice [29] and sows [39] in order to understand a potential interval of stress resilience and the gendered responses of the offspring. However, the mechanisms underlying maternal stress transmission to the fetus and stress buffering have been not investigated at the molecular level. Furthermore, potential genetic and epigenetic factors regulating the crosstalk between the skeletal and nervous system have been poorly explored [67]. A next important step could be to investigate which genes of the maternal and fetal HPA as well as uterine genes may be stress-sensitive with long-lasting effects leading to programmed adult bone diseases, compared to other protective factors (behavioral or nutritional) which may intervene on these presumed bone stress genes to rescue the environmental perturbations.

Of note, only two studies provided impressive but insufficient results of the genetic and epigenetic influences of ELS on bone deregulation [35,45]. The knowledge gap to mark neuronal plasticity and detrimental changes in bone metabolism and bone microstructure urges caution in determining a functional link between ELS and bone disease [20]. In fact, a growing body of evidence has characterized important epigenetic modifications affecting bone remodeling and metabolism during early and late bone development [68,69], also describing the progression of specific epigenetic pathways towards to a given adult bone disease, such as osteoporosis [19,20,21,70]. In this regard, an interesting interpretation was advanced by Weirtz-Kozak et al. [31] on the relationship between depressed individuals with experiences of childhood maltreatment and abuse and osteoporosis. Future studies should assess which factors might be consistent with particular developmental patterns of bone metabolic and inflammatory changes, associated with the vulnerability in adulthood to osteoporosis and fracture [20,43], but also musculoskeletal diseases, such as psoriatic arthritis [71]. There is a considerable need to characterize the epigenetic mechanisms potentially involved in the restoring of certain altered developmental trajectories of bone formation and body size caused by maternal stress as the primary origin of fetal and child defects [13]. In parallel, the identification of key epigenetic changes could act as ELS-predictive markers of developmental patterns during the gestational period, resulting in an early warning system of stress deterioration, to use a term coined by developmental instability research [38]. This recommendation also applies in the case of genetic bone disorders, such as infantile cortical hyperostosis (OMIM#114000), also called Caffey disease, and osteogenesis imperfecta (OMIM#166200), known as ‘‘battered child syndrome’’ or “brittle bone disease”, for which the use of a differential diagnosis is of paramount importance to help distinguish a suspected case of maltreatment from a bone pathology, as recently described [72,73]. The first step toward a preventive perspective of prenatal ELS has been recently provided by Verbruggen et al. [74] in evaluating biomechanical forces generated by fetal kicks and movements during the second half of gestation as a result of stimulation of the fetal skeleton related to the form of stress and strain. Authors, despite not including maternal stress surveys in their analysis, have discussed a potential link between fetal biomechanics and skeletal malformations using novel cine-magnetic resonance imaging technology to model fetal movements. Such findings stimulate future research to understand the biomechanical early warning signs, along with clinical, genetic, and epigenetic correlations, adopting a multi-integrative approach, by advanced molecular strategies and machine-learning applications [65].

## Figures and Tables

**Figure 1 children-10-00102-f001:**
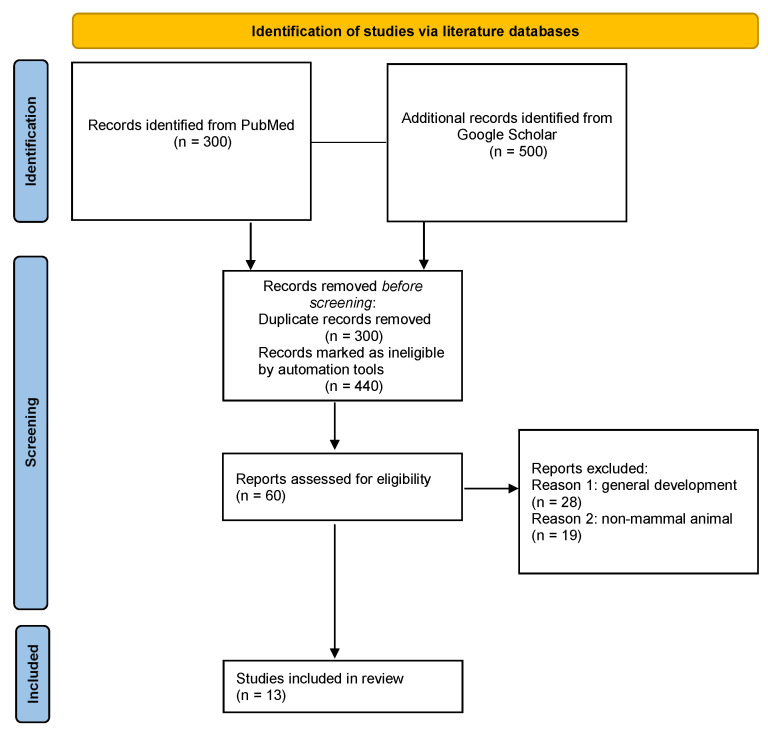
Schematic illustration of the PRISMA flowchart used in the present review.

**Table 1 children-10-00102-t001:** Main features analyzed in ELS studies of bone development included in the review.

Ref.	ELS Type	Stress Procedure	Stress Effect on Dams	Anti-Stress Effect on Dams	Skeletal	Gender	Catch-Up Features	Molecular Analysis
					Differences			
● Mouse								
[26]	Pre (GD1-16)	12 h of supine immobilization	Low maternal body weights during stress	no	Vertebral and sternal abnormalities, bipartite ossification; GR; embryos or fetus mortality.	no	no	no
[27]	Pre (GD7-9)	i.p. Dex (1 mg/kg or 10 mg/kg)	no	no	T13 or L1 vertebral anomalies.	Xiphoid process bigger in female embryos.	no	no
[28]	Pre (GD1-8)	6 h stuck in a restrainer	no	no	Low fetal body weight; altered number of somites, limb bud formation, regression of interdigital webbing.		no	Suppression of Igf1 and Acta1 genes; lower expression of Aldh1a2 and Fgf8.
[29]	Pre (GD12 to delivery)	45 min/3 times a day stuck in a restrainer	no	Chew a wooden stick	Lower bone mass, decrease in trabecular bone mass in both vertebrae and distal femur of the offspring of non-chewing stressed mothers	no	no	no
[30]	Pre (GD1-18)	Caloric restriction	no	no	Reduced skull size	no	no	no
[31]	Post (PND1-14)	Maternal separation during the dark cycle for 3 h	no	no	Low birth weight and altered bone innervation.	no	no	Altered neurogenic and osteogenic markers.
● Rat								
[32]	Pre (GD1-21)	45 min/3 times a day stuck in an immobilization bag	Less food consumption, weight loss	no	no	no	Faster growth and higher weight gain in offspring.	no
[33]	Pre (GD9-13)	i.m. Dex (100 micro g/kg)	no	no	no	Transient increases in crown–rump length and tibia and femur lengths at 3–6 weeks of age; altered cortical bone dimensions in 12-week-old female.	no	no
[34]	Pre (GD18 in F0 pregnant mice);	Bilateral uterine vessel ligation in F0 mothers at GD18;	no	no	no	Shortened femurs, reduced trabecular and cortical BMC in females (F1).	Low birth weight in F2 male and female offspring were postnatally recovered.	no
[35]	Pre (GD18-19 in F1 pregnant mice)	Physiological measurements (tail cuff blood pressure, glucose tolerance test, metabolic cage experiment) at GD18-19.						
[36]	Peri (GD14-21); (PND2-9)	light from 2000 h–0800 h, housing in a wire-mesh-floored pre-partum stress: cage (24 h), food deprivation (12 h), tilting the cage 45° (6 h), exposure to strobe light (1 h), forced swim (10 min), restraint (30 min), and wet bedding (10 h). Post-partum stress: strobe light, wet bedding, wire-mesh-floored cage, food deprivation, restraint, male in-truder (5 min), forced swim, and housing in a small cage.	no	no	no	Reduced tibia length.	Reduced femur length in males.	no
● Rabbit								
[37]	Pre (GD6-19)	Toxic compound: antiprotozoal agent (80, 320, and 1280 mg/kg)	no	Limb abnormalities in high-dose group	no	no	no	no
[38]	Pre (GD6-19)	Semi-toxic compound: hydroxy-propyl methylcellulose (100, 500, and 1500 mg/kg)	Less food consumption, weight loss		Low birth weight, transient alterations in ossifications	no	no	no
● Pig								
[39]	Pre (last 24 GD)	i.m. Dex (3 mg/sow 48 h dosage)	AKG (0.4 g/kg BW/day dosage)	no	Bone markers altered; less mineralized humeri, heavier weight in piglets of AKG-treated mothers.	no	no	no

Abbreviations: AKG (alpha-ketoglutarate); BMC (bone mineral content); Dex (Dexamethasone); GD (gestational day); GR: growth retardation; i.m. (intramuscular injection); i.p. (intraperitoneal injection); Peri (perinatal stress); Pre (prenatal stress); Post (postnatal stress); PND (postnatal day).

## Data Availability

Not applicable.

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
