# Peer review of "Early Life Stress (ELS) Effects on Fetal and Adult Bone Development"

_children, 2023, doi:10.3390/children10010102_

Round 1
Reviewer 1 Report
Dear Authors,
this is an interesting review of the literature on a significant topic.
One suggestion you may wish to consider, in making the your paper even better. You may wish to examine in the selected literature whether or not there exists a shift in methodological approaches as well as a shift in the research aims.
Author Response
Dear Children MDPI Editorial Team,
Thank you very much to the editorial committee and reviewers for considering our submission so carefully. We are pleased to submit a revised draft (children-2072465), which has been substantially improved as a result of comments received. All changes in the article have been highlighted in yellow.
We have addressed each of the raised points (Q: query; R: reply).
We hope that you will consider the new revised manuscript acceptable for publication.
Thank you for considering the manuscript again.
Kindest regards,
Prof. Enrico Parano
(enrico.parano@cnr.it)
- Response to Reviewer 1
Q1: You may wish to examine in the selected literature whether or not there exists a shift in methodological approaches as well as a shift in the research aims.
R1: We thank the reviewer for catching the point of our work, but we prefer not to lengthen the discussion paragraph since we stressed the heterogeneity of methodological approaches and research aims used in the examined literature that limits the quality of our analysis. However, we also compared the resolution power of achievements obtained from different animal models and stress schedule performed, also suggesting some new research proposals on this topic (Lines: 438-440; 450-518).
Reviewer 2 Report
Dear Author,
Thank you for the opportunity to review this article.
It is an interesting and novel topic in epigenetics. Far to have practical implications, this could be a cornerstone for the study of ELS.
Introduction is a bit hermetic and pretentious, it should be more easy to understand. “The relevance of ELS effects has been confirmed by short and long-term biological 38 changes ― both as structural and functional affected organs and tissues, attributed to 39 (mal-)adaptive genomic responses during the perinatal life that may lead to the program- 40 ming of adult diseases”. Can you give some practical examples of diseases? It is too synthetic for an orthopedic-themed paper. Or is it only for genetics?
Materials and methods are elaborate.
Prisma flowchart is not a result, it is merely a method. Meta-analysis graphs look completely different. Results should be rewritten to be relevant.
You name the Conclusion in the abstract, but you lack a Conclusion.
On rows 115-117 you emphasize the lack of a methodological approach to 115 identify and monitor the maternal factors that may negatively influence the fetal bone 116 mass and poor early growth. Here is an article that debates maternal risk factors for newborn fractures that we recommend you read and cite: Obstetric fractures in cesarean delivery and risk factors as evaluated by pediatric surgeons, published in Int Orthop, DOI 10.1007/s00264-022-05547-2.
This article should undergo a major revision before another attempt at publication.
Author Response
Dear Children MDPI Editorial Team,
Thank you very much to the editorial committee and reviewers for considering our submission so carefully. We are pleased to submit a revised draft (children-2072465), which has been substantially improved as a result of comments received. All changes in the article have been highlighted in yellow.
We have addressed each of the raised points (Q: query; R: reply).
We hope that you will consider the new revised manuscript acceptable for publication.
Thank you for considering the manuscript again.
Kindest regards,
Prof. Enrico Parano
(enrico.parano@cnr.it)
- Response to Reviewer 2
We would express our gratitude to reviewer for appreciating our task and for providing several constructive comments.
Q1: Introduction is a bit hermetic and pretentious, it should be more easy to understand. “The relevance of ELS effects has been confirmed by short and long-term biological 38 changes ― both as structural and functional affected organs and tissues, attributed to 39 (mal-)adaptive genomic responses during the perinatal life that may lead to the program- 40 ming of adult diseases”. Can you give some practical examples of diseases? It is too synthetic for an orthopedic-themed paper. Or is it only for genetics?
R1: According to your observation, we made the sentence more readable for the orthopedic-themed paper, also adding some didactive examples (Lines 42-70).
Q2: Materials and methods are elaborate. Prisma flowchart is not a result, it is merely a method. Meta-analysis graphs look completely different.
R2: Thank you so much for catching these confusing mistakes, which we have now corrected it. We simplified the methodological description (Lines 107-116) and we also removed the fig.1 from the result section reporting it in the methods section.
Q3: Results should be rewritten to be relevant.
R3: We eliminated any redundancy and made this section a bit more fluent and synthetic (Lines 132-34; 146-52; 158-73; 181-90; 195-209; 234-36; 262-65; 288-98; 304-08; 312-13; 345-47).
Q4: You name the Conclusion in the abstract, but you lack a Conclusion.
R4: We realized that the abstract should not be structured. We rewrite the abstract as a single paragraph without headings as you suggested (Lines: 11-24).
Q5: On rows 115-117 you emphasize the lack of a methodological approach to 115 identify and monitor the maternal factors that may negatively influence the fetal bone 116 mass and poor early growth. Here is an article that debates maternal risk factors for newborn fractures that we recommend you read and cite: Obstetric fractures in cesarean delivery and risk factors as evaluated by pediatric surgeons, published in Int Orthop, DOI 10.1007/s00264-022-05547-2.
R5: Thank for this tip. We considered and incorporated in the context this additional contribution to maternal influence on the offspring skeleton health (Lines 132-135; 497).
Q6: This article should undergo a major revision before another attempt at publication.
R6: Thank you for your assessment to improve the quality of the paper. As you may see, we entirely revised the text from the abstract to discussion.
Round 2
Reviewer 2 Report
Dear Author,
Thank you for your consideration.
We carefully read your paper in the revised edition.
The manuscript packs a lot of information and it needs careful reading in order to assimilate it.
Your article needs a bit of English revision, but it gathers more structure now.
Only a minor English revision is needed prior to being published.